# Effect of Individualized Oral Health Care Training Provided to 6–16-Year-Old Psychiatric In-Patients—Randomized Controlled Study

**DOI:** 10.3390/ijerph192315615

**Published:** 2022-11-24

**Authors:** Benedikt Bock, Arndt Guentsch, Roswitha Heinrich-Weltzien, Christina Filz, Melanie Rudovsky, Ina M. Schüler

**Affiliations:** 1Section of Preventive and Paediatric Dentistry, Jena University Hospital, 07743 Jena, Germany; 2Department of Surgical Sciences, Marquette University School of Dentistry, Milwaukee, WI 53201-1881, USA; 3Department of Child and Adolescent Psychiatry, Psychotherapy and Psychosomatics, Jena University Hospital, 07743 Jena, Germany

**Keywords:** toothbrushing, dental plaque, child, adolescent, mental disorders, motivational interview, oral health

## Abstract

Background: To assess the effect of individualized oral health care training (IndOHCT) administered to 6–16-year-old psychiatric in-patients on dental plaque removal. Methods: 74 in-patients with mental health disorders (49 males) aged 6–16 years with a mean age of 10.4 ± 2.3 years, were randomly divided into two equal groups. At the start of hospitalization, one calibrated dentist assessed the oral health status in the hospital setting. In-patients of the intervention group (IG) received IndOHCT, while those of the control group (CG) got an information flyer. Dental plaque was assessed by the Turesky modified Quigley-Hein-Index (TI) at the start (t0) and at the end of hospitalization before (t1a) and after (t1b) autonomous tooth brushing. Results: During hospitalisation, the TI was reduced in both groups (t0→t1a: IG = −0.1; CG = −0.2, *p* = 0.71). However, in-patients receiving IndOHCT achieved significantly higher plaque reduction rates than the controls when plaque values before and after autonomous tooth brushing were compared (t1a→t1b: IG = −1.0; CG = −0.8; *p* = 0.02). The effect size (ES) demonstrates the efficacy of IndOHCT (ES = 0.53), especially in children with mixed dentition (ES = 0.89). Conclusions: IndOHCT enabled hospitalized children and adolescents with mental health disorders to achieve a better plaque reduction by tooth brushing but failed to improve self-controlled routine oral hygiene.

## 1. Introduction

Patients with mental health disorders might be more susceptible to oral health problems than the general population due to several risk factors: a lack of motivation for self-care and oral hygiene, xerostomia caused by psychiatric medications, as well as fear and difficulty in accessing oral healthcare services [1,2,3]. Concomitantly, dentists experience high levels of their own psychological distress when treating children or adolescents with psycho-emotional disorders [4]. Children with mental health disorders are 85–93% more likely to experience oral health problems compared to those without mental health disorders [5]. Furthermore, adverse childhood experiences, like child abuse and neglect, parental divorce, domestic violence, caregiver mental illness, exposure to drug abuse, or struggle with family income, were identified as toxic stressors and associated with poor oral health in children and adolescents [6,7]. There is evidence that adults with mental health disorders are at a higher risk of caries, periodontal disease, and poor oral hygiene than the general population [8,9]. Since mental health problems and disorders occurring during childhood often persist in adulthood [10], intensive preventive oral health care in children and adolescents with psychiatric disorders might reduce the burden of oral diseases in adults.

In Europe, 9.9% of all schoolchildren require mental health care [11]. In Germany, the published prevalence of psycho-emotional disorders ranges between 8.6% [12], 16.9% [13], and 28% [14]. Despite this considerable prevalence, literature regarding oral health in children and adolescents with mental health disorders is scarce and targeted mainly on those with attention deficit hyperactivity disorder (ADHD) and autism spectrum disorder (ASD). Inconsistent data is available about children and adolescents with ADHD, revealing a higher caries prevalence [15,16,17,18], a higher prevalence of gingivitis [16], no differences in dmft/DMFT Index [19], but more neglected oral hygiene [17,19] than peers without ADHD. Furthermore, children and adolescents with ASD have a higher prevalence of caries and gingivitis [20,21,22] due to neglected oral hygiene compared to healthy peers.

Dental plaque reduction has consistently been recognized as an indispensable element of preventing gingivitis and dental caries [23]. The recommendation of regular tooth brushing with fluoridated toothpaste at least twice daily as a basic preventive measure is based on extensive evidence [24]. Nevertheless, the reliable execution of meticulous oral hygiene cannot always be assured, especially in hospitalized children [25,26]. In a hospital setting, the focus on the patients’ primary medical condition might overshadow oral health care needs [27]. However, hospitalization also provides the opportunity to reach children and adolescents at risk for poor oral health. In this setting, oral health education and guidance might be provided individually within a controlled and trustworthy environment.

Oral health education focussing solely on oral health literacy has been demonstrated to have little effect on improving oral health or oral health behaviour [28]. However, motivational interviewing techniques increased that effect [29,30,31]. More pragmatic approaches focusing on improving individual tooth brushing skills by guided training were implemented successfully [32]. Therefore, the oral health care training provided to children and adolescents hospitalized with psychiatric disorders was individualized and focussed to the specific personal oral health conditions. Priority was given to practically train tooth-brushing skills in a motivational approach.

The aim of this study was to assess the effect of the individual oral health care training (IndOHCT) given to 6–16-year-old in-patients hospitalized with mental health disorders on dental plaque removal. Three working hypotheses were tested. 1. The oral hygiene status of child and adolescent in-patients with mental health disorders receiving IndOHCT at the start of hospitalization is better at the end of hospitalization than the oral hygiene of those not trained. 2. Child and adolescent in-patients with mental health disorders receiving IndOHCT achieve higher plaque reduction rates by tooth brushing compared to those not trained. 3. The efficacy of IndOHCT is independent of the patient’s particular psychiatric diagnosis.

## 2. Materials and Methods

This randomized unblinded clinical controlled study was conducted from 05/2013 to 11/2014 at the Department of Child and Adolescent Psychiatry, Psychotherapy, and Psychosomatics, Jena University Hospital, Germany.

### 2.1. Study Population

Study-participation was open to all newly admitted in-patients aged 6–17 years and offered at hospital admission. Eighty-one children agreed to participate in this study. The dropout rate was 8.6% (*n* = 7). One patient refused further participation after initial consent and six patients left the hospital prematurely. Finally, 74 hospitalized children and adolescents aged six to 16 years (male = 49, mean age: 10.4 ± 2.3 years) with mixed and permanent dentitions were included, amounting to 21.7% of all new admitted in-patients during the study period (*n* = 341), (Figure 1).

In-patients were divided into two equal groups by a simple randomization list generated by a statistician: an intervention group (IG) and a control group (CG).

A sample size calculation was performed to estimate the number of in-patients necessary to observe a significant plaque reduction after IndOHCT compared to a non-trained control group with 80% power and alpha = 0.05 tested by a two-sided *t*-test. It was estimated that plaque reduction by 1.0 point of the Turesky index is clinically relevant. The required sample size per group was 37 to prove the superiority of the IndOHCT. Statistical analysis of the disorder subgroups was performed if a minimum of five in-patients were available for inclusion.

The statistical analysis included those patients whose data could be collected at all three assessment time points. In-patients were enrolled until the estimated sample size was reached.

### 2.2. Medical Data Collection

Mental disorders and medications of the in-patients were collected from the medical records after the completion of the study. Mental health disorders were diagnosed according to the multiaxial diagnostic pattern of the international statistical classification of diseases and health related problems (ICD-10) and the diagnostic and statistical manual of mental disorders (DSM-IV). In-patients with more than one mental health disorder or medication were included in each applicable disorder subgroup.

### 2.3. Oral Examinations

All in-patients were orally examined within the first week after admission to the hospital and three to four days before dismissal from the hospital. The examinations were performed by one dentist (BB) using a standard ball-end probe and an illuminated mirror (DenLite©, Miltex, Skaneateles Falls, NY, USA) between 8:30 and 11:00 am in a separate examination room at the hospital. The in-patients sat on a chair or examination bed. Teeth were dried with cotton rolls. No radiographs were available.

Dental plaque was scored by the Turesky modified Quigley-Hein Index (TI) [33] after staining all surfaces of the teeth with Rondells Blue© (Directa, Upplands Väsby, Sweden). Caries was recorded by the dmft/DMFT index [34] and gingival health by the periodontal screening index (PSI) [35] after tooth brushing. The TI was determined three times: at the first oral examination after admission to hospital before tooth brushing (t0), and two times at the second examination before dismissal from the hospital, and before (t1a) and after autonomous tooth brushing (t1b). The mean time span between the first and second examination was 23.2 ± 6.9 days.

This studie’s primary outcome parameter was the difference of TI values between the IG and CG for two situations: (a) at the start and at the end of hospitalization and (b) before and after autonomous tooth brushing. The prevalence and experience of dental caries and gingivitis were recorded to characterize the oral health status of the study population.

Before the examinations, the examiner (BB) was calibrated by an experienced dentist and epidemiologist (IMS). The inter- and intra-examiner reproducibility of TI values were assessed with kappa (k) statistics. The inter-examiner reproducibility was 0.92. The intra-examiner reproducibility were 0.95 (BB) and 0.97 (IMS).

### 2.4. Intervention

In-patients of the IG received IndOHCT after the first oral examination. This training included three elements: oral health information, motivation towards oral health benefits, and practical tooth brushing exercises. Information provided included the patient’s individual oral health and oral hygiene status, as well as the aetiology and consequences of dental plaque, gingivitis, and caries. The reason for the recommended frequency and duration of tooth brushing was explained. The time spent on conversation amounted to approximatively 10–15 min. After the plaque revelation with the staining agent, the dentist first observed the in-patients while brushing their teeth as they usually do without interference. Then, with help of the dentist, the in-patient discovered insufficiently cleaned tooth surfaces, visible due to the remaining staining agent. For those surfaces, alternative brushing methods were explored, tested, and trained. This IndOHCT was conducted in a collaborative, patient-centred, and goal directed form with the aim to elicit and strengthen the in-patients´ intrinsic motivation for change. Strengths and deficiencies in teeth brushing techniques were reflected in an age-related conversation, questions were explored, advice was given, and goals were set.

All in-patients of the CG received an identical information flyer [36] after the oral examination. Neither counselling nor tooth brushing training were provided.

During hospitalization, all in-patients performed self-controlled oral hygiene. No oral hygiene related instructions were given to the medical staff, nurses, or parents/legal guardians.

### 2.5. Data Analysis

The *t*-test for the analysis of mean values and the Fisher´s exact test for categorical data were applied using SPSS for Windows (Statistical Package for Social Science, Version 22.0). The effect size (Cohen´s d) was calculated to demonstrate the effect of IndOHCT on plaque reduction and was categorized as small (≤0.4), medium (0.5–0.7), or large effect size ES (≥0.8) [37]. The level of significance was set at *p* ≤ 0.05 and no correction for multiple analyses was applied.

## 3. Results

### 3.1. Demographic Data

Table 1 presents the demographic characteristics age, gender, and dentition in the study population. All children and adolescents had at least one permanent tooth.

### 3.2. Oral Health

Table 2 displays the oral health parameters of the study population observed at the first examination. Children and adolescents hospitalized with mental health disorders demonstrated a higher caries prevalence and experience in the primary dentition than in the permanent dentition, while gingivitis prevalence and dental plaque scores were higher in the permanent than in the primary dentition.

Untreated dental caries was detected in two thirds of the primary dentitions and more than half of the permanent dentitions. More than half of all in-patients revealed gingivitis. Periodontitis was not diagnosed in this cohort. There were no significant differences between IG and CG in the prevalence and experience of dental caries, gingivitis, and dental plaque.

### 3.3. Individualized Oral Health Care Training (IndOHCT)

Regardless of receiving the IndOHCT or not, both groups reduced their plaque values at t1a in comparison to the baseline findings in both dentitions, but these reductions did not reach statistical significance (Table 3).

Within the subgroups of psychiatric diagnoses and medication, there were also no statistically significant differences in TI between IG and CG, except for in-patients with abnormal educational conditions at the baseline (Table 4).

The effect of IndOHCT on tooth brushing efficacy was determined by comparing plaque values before (t1a) and after (t1b) autonomous tooth brushing. In-patients of the IG reached higher plaque reduction rates than those of the CG (*p* = 0.02) in both dentitions (Table 5). The overall effect size of IndOHCT for plaque reduction was 0.53 (medium effect). There was a large effect (ES = 0.89) on younger in-patients with mixed dentition and a very small effect (ES = 0.04) on older in-patients.

A subgroup analysis revealed differences in plaque reduction rates depending on the kind of mental health disorder (Table 6). In in-patients with the diagnosis “acute, stressful life events” or ADHD medication, the IndOHCT induced significantly better plaque reduction with the highest effect. In-patients with “behavioural and emotional disorders with onset usually occurring in childhood and adolescence”, “mental disorder, deviant behaviour or disability in the family” and “abnormal immediate surroundings” reached significantly better tooth cleaning after IndOHCT with a medium effect.

A closer look at the different plaque localizations revealed that in-patients of the IG were superior to those of the CG with regard to cleaning of posterior teeth, left side of the dentition, and smooth and proximal surfaces (Table 7). In both groups, IG and CG, the plaque reduction was higher on the anterior teeth than on the posterior teeth and the vestibular surfaces were cleaned better than the oral surfaces. No differences were found in the comparison between the right and left sides of the dentition or between the smooth surfaces and the approximal surfaces of the teeth.

## 4. Discussion

This randomized clinical trial investigated the effect of IndOHCT in children and adolescents hospitalized with mental health disorders. The assessed oral health parameters dental caries and gingivitis were considerable higher than those reported in germen children and adolescents with intellectual disabilities [38]. This finding emphasises the need for improved oral care in this group of special needs patients.

The first hypothesis has to be rejected. IndOHCT failed to improve the oral hygiene status during hospitalization, although the training was based on the information-motivation-behavioural skills model [39] and the motivational interviewing approach, which are reported as having potential to help patients with poor oral health [30,31].

Tooth brushing, a simple motor activity proven to enhance oral health, may over time become a less conscious action. Existing habits can inhibit new learning and exert proactive interference on learning new skills [40]. It is generally accepted that strategies to promote changes in oral health behaviour should be based on behavioural theory [41]. Therefore, the training intervention conducted in this study was designed in accordance with the information-motivation-behavioural skills model [39].

With the aim to enhance the in-patients’ tooth brushing skills and behaviour, the dentist communicated information which is the first element of the information-motivation-behavioural skills model.

At first the dentist explained, in age-specific language, the patient’s individual oral health status regarding caries, gingivitis, and plaque and how correct tooth brushing can help to prevent them. Furthermore, he observed and communicated individual brushing deficiencies. The in-patients participated actively in exploring and testing new methods to empower themselves to achieve better oral hygiene.

Secondly, the dentist promoted motivation to learn and train the personalized, adequate tooth brushing technique and encouraged a positive attitude towards performing proper tooth brushing twice daily.

Thirdly, he ensured that all steps of the behavioural skill of tooth brushing, including having time and space to brush, the required supplies, and self-efficacy to perform the task correctly, were understood and that they could be performed autonomously.

All elements of the information-motivation-behavioural skills model were implemented and techniques of motivational interviewing [29] were applied. Care was taken to communicate and act age-related, since major differences in cognitive and motor skills in the age range from 6–16 years were to be respected. Although the information-motivation-behavioural skills model and motivational interviewing techniques were implemented, the IndOHCT failed to improve the oral hygiene status during hospitalization. In the literature, oral hygiene was observed to be neglected in children during hospitalization compared to home care [25]. Hospitalization involves unfamiliar surroundings, different circadian cycles, medical therapy, and worries about the medical conditions. Familiar tooth brushing habits have to be implemented in this unfamiliar setting. Medical needs are a matter of priority and compete with less visible oral health care needs [27]. Nevertheless, in the present study, awareness of oral health was raised among paediatric nurses as a side effect of conducting oral examinations and IndOHCT in the hospital setting. Oral health is often neglected by health professionals and caregivers in favour of the general symptomatology that the in-patients present during the hospital stay [42]. It is therefore possible that the nursing staff, without knowledge about the assignment of in-patients into groups, started to deliberately control the execution of tooth brushing. This interference might have biased the outcome of plaque reduction during hospitalization by reducing differences between groups.

However, patients receiving IndOHCT were able to brush their teeth more effectively by achieving higher plaque reduction rates during autonomous tooth brushing, compared to those not trained, confirming the second hypothesis, especially in children with mixed dentitions.

This finding is in line with observations regarding practical tooth brushing training in first graders, were a pragmatic approach emphasizing improving individual tooth brushing skills was successful in increasing the areas of brushed tooth surfaces [32].

The efficacy of IndOHCT differed between different mental health disorders, infirming the third hypothesis. In-patients with “acute, stressful life events” and ADHD medication benefited most from IndOHCT, with the largest effect sizes observed in both of these groups. All other in-patient groups profited from the training intervention with a medium effect size.

In-patients coping with “acute, stressful life events” and those receiving ADHD medication might have been more susceptible to perceiving the IndOHCT as caring attentiveness, emotionally positive distraction from their daily problems, and empowerment to do something for their own health and well-being than the in-patients with other mental health disorders.

The highest dental plaque levels were observed in in-patients receiving ADHD medication, which confirms findings from other studies [18,19,43]. In children and adolescents with ADHD, cognitive and executive functions may be impaired, leading to significant behavioural problems that affect everyday life [44], including poorer tooth brushing habits [19,45].

Based on the present results, children and adolescents with mental health disorders profit from IndOHCT at a hospital setting. Trained in-patients were superior to the untrained ones in effective tooth cleaning.

### Strengths and Limitations

The present study contributes to the evidence that IndOHCT is effective for enhancing tooth brushing skills in children and adolescents hospitalised with mental health disorders. The used information flyer was identical for all age groups, so it may not have been equally appropriate for all in-patients, especially the younger ones. The level of intelligence influences cognition and knowledge acquisition, especially in children and adolescents with mental health disorders. Training programs must be adapted or specifically designed for individuals with reduced intelligence. No cases of major intelligence impairment occurred in our study cohort. Therefore, the study results are not generalizable to children with mental disorders and reduced intelligence. The effect size was calculated to demonstrate the magnitude of differences between groups. Efforts were taken to reduce selection bias by randomised selection. Only one calibrated examiner performed all measurements to limit observation bias, but a lack of blinding might have increased the risk of observation bias. All included in-patients were living in central Germany. This limited provenance strengthens the homogeneity and comparability between groups but reduces the external validity and generalizability. The sample size was estimated to evaluate the effect of IndOHCT on plaque reduction, and therefore comparisons between subgroups of psychiatric diagnoses, medications and plaque loci are statistically underpowered. These results should be interpreted with caution.

## 5. Conclusions

IndOHCT, conducted by the information-motivation-behavioural skills model, enabled hospitalized children and adolescents with mental health disorders to have better plaque reduction by tooth brushing, especially those with mixed dentitions, although it failed to improve self-controlled routine oral hygiene during hospitalization. The high prevalence of psycho-emotional problems in children and adolescents associated with a high prevalence of caries and gingivitis demands a focus on individual preventive measures, including individualized oral healthcare training toward enhancing practical tooth brushing skills. Further efforts are necessary to improve the daily oral hygiene behaviour in hospitalized children and adolescents.

## Figures and Tables

**Figure 1 ijerph-19-15615-f001:**
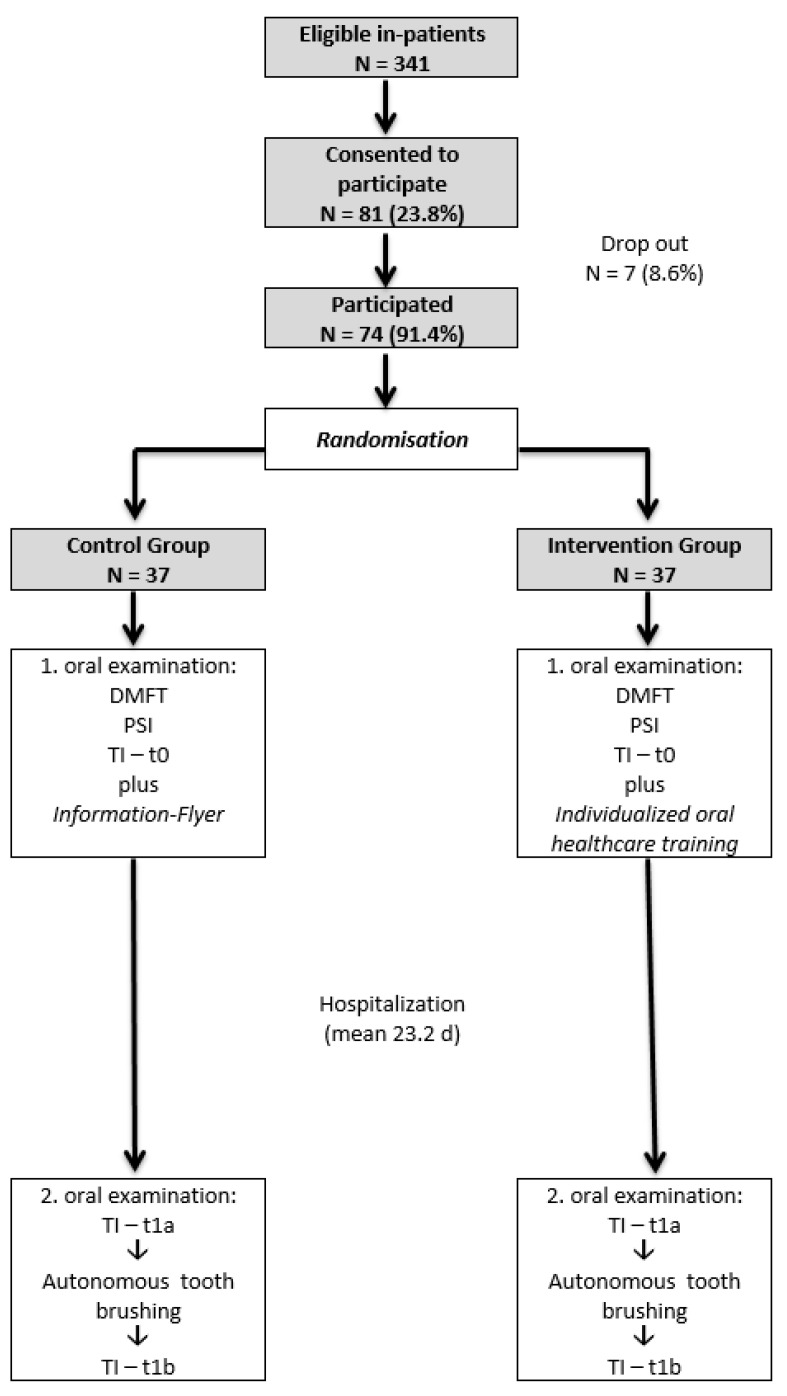
Study design.

**Table 1 ijerph-19-15615-t001:** Demographic characteristics of the study population, intervention group (IG), and control group (CG).

Demographical Characteristics	IG	CG	All
Male (n, %)	23 (62.2%)	26 (70.3%)	49 (66.2%)
Female (n, %)	14 (37.8%)	11 (29.7%)	25 (33.8%)
Age in years (Mean ± SD)	10.7 (2.5)	10.2 (2.0)	10.4 (2.3)
In-patients aged 6–11 years (n, %)	22 (59.4%)	26 (70.3%)	48 (64.9%)
In-patients aged 12–16 years (n, %)	15 (40.5%)	11 (29.7%)	26 (35.1%)
In-patients with mixed dentition (n, %)	20 (54.1%)	26 (70.3%)	46 (62.2%)
In-patients with permanent dentitions (n, %)	17 (45.9%)	11 (29.7%)	28 (37.8%)

**Table 2 ijerph-19-15615-t002:** Oral health in child and adolescent psychiatric in-patients at the beginning (t0).

	** *n* ** **Patients** **(%)**	**Caries** **Prevalence (%)** **dmft > 0** **(95% CI)**	**Untreated Caries Prevalence (%)** **d > 0** **(95% CI)**	**dmft** **(SD)**	**dt** **(SD)**	**Gingivitis** **Prevalence (%)** **PSI > 0** **(95% CI)**	**PSI** **(SD)**	**Dental Plaque** **TI at t0** **(SD)**
Primary teeth	46 (100.0)	84.8 (73.9–93.5)	67.4 (54.4–80.4)	3.6 (2.8)	1. 8 (2.0)	50.0 (34.8–65.2)	0.2 (0.2)	1.9 (0.5)
CG	26 (56.5)	88.5 (75.0–100.0)	65.4 (45. 5–83.9)	4.0 (3.1)	1.8 (2.3)	53.9 (34.8–71.0)	0.2 (0.2)	2.0 (0.5)
IG	20 (43.5)	80.0 (60.0–95.2)	70.0 (47.6–89.5)	3.1 (2.4)	1.8 (1.5)	45.0 (23.5–66.7)	0.2 (0.2)	1.8 (0.5)
	** *n* ** **Patients** **(%)**	**Caries** **prevalence (%)** **DMFT > 0** **(95% CI)**	**Untreated caries** **prevalence (%)** **D > 0** **(95% CI)**	**DMFT** **(SD)**	**DT** **(SD)**	**Gingivitis** **prevalence (%)** **PSI > 0** **(95% CI)**	**PSI** **(SD)**	**Dental plaque** **TI at t0** **(SD)**
Permanent teeth	74 (100.0)	60.8 (50.0–71.6)	56.8 (44.6–68.9)	2.3 (2.8)	1.7 (2.2)	58.1 (47.3–68.9)	0.2 (0.2)	2.3 (0.5)
CG	37 (50.0)	62.2 (46.9–77.8)	56.8 (41.7–71.8)	2.4 (2.6)	1.7 (2.2)	59.5 (43.6–75.6)	0.2 (0.2)	2.4 (0.5)
IG	37 (50.0)	59.5 (40.6–75.0)	56.8 (41.4–72.2)	2.2 (3.0)	1.7 (2.3)	56.7 (41.5–73.0)	0.2 (0.2)	2.2 (0.5)

**Table 3 ijerph-19-15615-t003:** Dental plaque in child and adolescent psychiatric in-patients (IG vs. CG) at the beginning (t0) and the end of hospitalization (t1a) with mixed and permanent dentitions.

In-Patients	*n* Patients CG/IG	TI at t0				TI at t1a			
		All Mean (SD)	CG Mean (SD)	IG Mean (SD)	*p*	All Mean (SD)	CG Mean (SD)	IG Mean (SD)	*p*
All	37/37	2.2 (0.5)	2.3 (0.5)	2.1 (0.5)	0.22	2.0 (0.5)	2.1 (0.5)	2.0 (0.5)	0.39
In-patients with mixed dentition	26/20	2.2 (0.5)	2.2 (0.5)	2.1 (0.5)	0.35	2.0 (0.4)	2.1 (0.5)	2.0 (0.4)	0.53
In-patients with permanent dentition	11/17	2.2 (0.4)	2.3 (0.4)	2.2 (0.5)	0.34	2.0 (0.6)	2.1 (0.6)	2.0 (0.5)	0.56

**Table 4 ijerph-19-15615-t004:** Dental plaque in child and adolescent psychiatric in-patients (IG vs. CG) at the beginning (t0) and the end of hospitalization (t1a) in different mental disorder groups.

Psychiatric Diagnosis, Medication	*n* Patients CG/IG	TI at t0				TI at t1a			
		All Mean (SD)	CG Mean (SD)	IG Mean (SD)	*p*	All Mean (SD)	CG Mean (SD)	IG Mean (SD)	*p*
Neurotic, stress-related, and somatoform disorders (ICD 10—F40-F48)	13/13	2.2 (0.5)	2.3 (0.5)	2.2 (0.5)	0.36	2.0 (0.5)	2.0 (0.5)	2.0 (0.4)	0.86
Behavioural and emotional disorders with onset usually occurring in childhood and adolescence (ICD 10—F90-F98)	33/31	2.2 (0.5)	2.3 (0.5)	2.2 (0.5)	0.41	2.1 (0.4)	2.1 (0.5)	2.0 (0.4)	0.32
Abnormal intrafamilial relations (DSM-IV Axis 5-1)	11/14	2.3 (0.4)	2.3 (0.3)	2.2 (0.5)	0.47	2.0 (0.6)	2.0 (0.5)	2.0 (0.6)	0.72
Mental disorder, deviant behaviour, or disability in the family (DSM-IV Axis 5-2)	24/20	2.2 (0.5)	2.3 (0.5)	2.1 (0.5)	0.15	2.0 (0.5)	2.1 (0.5)	2.0 (0.4)	0.47
Abnormal educational conditions (DSM-IV Axis 5-4)	20/26	2.3 (0.4)	2.4 (0.4)	2.2 (0.4)	0.05 *	2.2 (0.4)	2.2 (0.4)	2.1 (0.1)	0.20
Abnormal immediate surroundings (DSM-IV Axis 5-5)	30/30	2.2 (0.5)	2.3 (0.5)	2.1 (0.5)	0.23	2.0 (0.5)	2.1 (0.5)	2.0 (0.4)	0.31
Acute, stressful life events (DSM-IV Axis 5-6)	12/10	2.3 (0.4)	2.3 (0.3)	2.3 (0.4)	0.85	2.0 (0.4)	2.0 (0.4)	2.1 (0.4)	0.65
Attention deficit hyperactivity disorder (ADHD) medication	9/11	2.4 (0.5)	2.4 (0.6)	2.3 (0.4)	0.57	2.0 (0.4)	2.2 (0.5)	1.9 (0.3)	0.19
Antipsychotic drug medication	7/5	2.3 (0.6)	2.3 (0.6)	2.3 (0.6)	0.88	2.2 (0.5)	2.4 (0.5)	2.0 (0.2)	0.18

* significant (*t*-test between IG and CG).

**Table 5 ijerph-19-15615-t005:** Dental plaque reduction rates in IG and CG after autonomous tooth brushing in the mixed and permanent dentition and effect size of IndOHCT.

In-Patients	*n* Patients CG/IG	TI—Difference t1a–t1b			Effect Size
		CG Mean (SD)	IG Mean (SD)	*p*	
All	37/37	−0.8 (0.3)	−1.0 (0.3)	0.02 *	0.53
In-patients with mixed dentition	26/20	−0.8 (0.32)	−1.1 (0.3)	0.01 *	0.89
In-patients with permanent dentition	11/17	−0.9 (0.3)	−0.9 (0.3)	0.94	0.04

* significant (*t*-test between IG and CG).

**Table 6 ijerph-19-15615-t006:** Dental plaque reduction rates in IG and CG after autonomous tooth brushing in mental disorder groups and effect size of IndOHCT.

Psychiatric Diagnosis, Medication	*n* Patients CG/IG	TI—Difference t1a–t1b			Effect Size
		CG Mean (SD)	IG Mean (SD)	*p*	
Neurotic, stress-related, and somatoform disorders (ICD 10—F40-F48)	13/13	−0.8 (0.3)	−1.0 (0.3)	0.08	0.74
Behavioural and emotional disorders with onset usually occurring in childhood and adolescence (ICD 10—F90-F98)	33/31	−0.8 (0.3)	−1.0 (0.3)	0.02 *	0.59
Abnormal intrafamilial relations (DSM-IV Axis 5-1)	11/14	−0.8 (0.3)	−1.0 (0.4)	0.19	0.56
Mental disorder, deviant behaviour, or disability in the family (DSM-IV Axis 5-2)	24/20	−0.8 (0.3)	−1.0 (0.3)	0.05 *	0.58
Abnormal educational conditions (DSM-IV Axis 5-4)	20/26	−0.8 (0.4)	−1.0 (0.3)	0.10	0.56
Abnormal immediate surroundings (DSM-IV Axis 5-5)	30/30	−0.8 (0.3)	−1.0 (0.3)	0.02 *	0.63
Acute, stressful life events (DSM-IV Axis 5-6)	12/10	−0.7 (0.2)	−1.0 (0.3)	0.02 *	1.01
Attention deficit hyperactivity disorder (ADHD) medication	9/11	−0.8 (0.3)	−1.1 (0.2)	0.04 *	0.96
Antipsychotic drug medication	7/5	−1.0 (0.3)	−1.0 (0.1)	0.66	0.58

* significant (*t*-test between IG and CG).

**Table 7 ijerph-19-15615-t007:** Dental plaque reduction rates in IG and CG after autonomous tooth brushing at different loci.

Locus	TI—Difference t1a–t1b				
	CG Mean (SD)	*p*	IG Mean (SD)	*p*	*p* CG vs. IG
Anterior teeth	−0.9 (0.4)	0.00 *	−1.1 (0.4)	0.01 *	0.09
Posterior teeth	−0.7 (0.3)		−0.9 (0.3)		0.02 *
Right side	−0.8 (0.4)	0.86	−0.9 (0.3)	0.38	0.10
Left side	−0.8 (0.3)		−1.0 (0.3)		0.01 *
Vestibular surfaces	−1.1 (0.5)	0.00 *	−1.3 (0.5)	0.00 *	0.08
Oral surfaces	−0.5 (0.3)		−0.7 (0.3)		0.08
Smooth surfaces	−0.8 (0.3)	0.98	−1.0 (0.3)	0.90	0.03 *
Approximal surfaces	−0.8 (0.3)		−1.0 (0.3)		0.02 *

* significant (*t*-test between IG and CG).

## Data Availability

The datasets used and analysed during the current study are available from the corresponding author on request.

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
