# Peer review of "Effect of Individualized Oral Health Care Training Provided to 6–16-Year-Old Psychiatric In-Patients—Randomized Controlled Study"

_ijerph, 2022, doi:10.3390/ijerph192315615_

Round 1

Reviewer 1 Report

Very good paper with good methodology and research design. The aim of this study was to assess the effect of individual oral health care training  (IndOHCT) given to 6-16 year old in patients hospitalized with mental care training on dental plaque removal in Germany. The subject is very intresting and relevant in the field. The high prevelence of psycho-emotional problems in children and adolescents associated with high prevalence of caries and gingivitis demands a focus on individual preventive measures. The methodology is good. Should have a larger sample size. The references are appropriate.

Author Response

General remarks of the authors

We are grateful for the comments of the two reviewers. All comments were carefully considered and incorporated into this new draft of the manuscript. The feedback allowed us to clarify important aspects of the study. We are convinced that the revised manuscript presented herewith is improved over the previous version. This letter will address all comments made in full detail and explain the respective changes we have made to the text.

Reply to reviewer 

Concern of the reviewer:

Very good paper with good methodology and research design. The aim of this study was to assess the effect of individual oral health care training  (IndOHCT) given to 6-16 year old in patients hospitalized with mental care training on dental plaque removal in Germany. The subject is very intresting and relevant in the field. The high prevelence of psycho-emotional problems in children and adolescents associated with high prevalence of caries and gingivitis demands a focus on individual preventive measures. The methodology is good. Should have a larger sample size. The references are appropriate.

Our response:

Thank you very much for this appreciating comment. We also believe in the importance of reaching this group of vulnerable patients in order to improve their weaker oral health.
Regarding the sample size, we did in advance a sample size estimation in cooperation with a statistician. Targeting at the reduction of TI, the necessary sample size to reach 80% power and alpha 0.05 was 37. A larger sample size would have increased the power of the subgroup analyses.

Reviewer 2 Report

-The words "insignificantly reduced" in the abstract is not clear.

- How could patient of 6-17 years old decide to retrieve consent? [Ref. 2.1, line 3]

- Why mental disorder and medication data was collected after completion of study? Do the authors think there could have been ethical concerns with that especially if the condition or medication was of any serious concern effecting the data collection or selection of the participant? If so, how was this dealt? [Ref. 2.2 line-1]

- How was random allocation done?

-How could mental compromised/ psychiatric patients that too children would understand or get motivated to the information given through pamphlet?

Author Response

General remarks of the authors

We are grateful for the comments of the two reviewers. All comments were carefully considered and incorporated into this new draft of the manuscript. The feedback allowed us to clarify important aspects of the study. We are convinced that the revised manuscript presented herewith is improved over the previous version. This letter will address all comments made in full detail and explain the respective changes we have made to the text.

The information Flyer was uploaded as supplimental material.

Reply to Reviewer 

  1. Concern of the reviewer:

The words "insignificantly reduced" in the abstract is not clear.

Our response:

Thank you for this remark. The TI between baseline (t0) and t1a was reduced, but this reduction was not statistically significant. We deleted the word “insignificantly” from this sentence. By including the p-value, we report the missing statistical significance of those reductions.

                Revised Text:

During hospitalisation, the TI was insignificantly reduced in both groups (t0àt1a: IG=-0.1; CG=-0.2, p=0.71).

  1. Concern of the reviewer:

 How could patient of 6-17 years old decide to retrieve consent? [Ref. 2.1, line 3]

Our response:

Thank you for this question. We requested written informed consent from all patients and their legal guardians at study enrolment.  Unwillingness to continue participation, expressed by either inpatient or legal guardian, was considered reason to exclude. One teen-aged patient refused further participation after initial consent (given by parents and herself).
We modified this sentence for more clarification.

Revised text:

One patient decided to retrieve consent refused further participation after initial consent and 6 six patients left the hospital prematurely.

  1. Concern of the reviewer:

Why mental disorder and medication data was collected after completion of study? Do the authors think there could have been ethical concerns with that especially if the condition or medication was of any serious concern effecting the data collection or selection of the participant? If so, how was this dealt? [Ref. 2.2 line-1]

Our response:

In preparation for this study, we agreed with our psychiatry colleagues and co-authors that we would not formulate inclusion or exclusion criteria based on psychiatric diagnoses. It should be possible to include all hospitalized children and adolescents. Only refusal to participate (expressed by the inpatient and their legal guardians) limited the study population.
After completing the oral assessments and the intervention, medical data was extracted from the medical records and added in the database. This way, the dental examiner was blinded to the specific mental disorder of each patient. This blinding allowed for greater impartiality of the examiner toward the patient. During the interview, the examiner focused on the patient's current abilities and willingness to cooperate – without knowledge of the specific mental disorder.

Revised text:

none

  1. Concern of the reviewer:

How was random allocation done?

Our response:

Before enrolment of the first inpatient, the statistician had created a simple randomization list for the patient number. Patient numbers were allocated in the order of admission. Conforming to this list, the patients were assigned to both groups. No selection was made according to age or sex. This way of randomization led to groups with the following demographic characteristics.

Demographic details:

Demographic Data

IG

CG

All

Male (n, %)

 23 (62,2%)

 26 (70,3%)

 49 (66,2 %)

Female (n, %)

 14 (37,8%)

 11 (29,7%)

 25 (33,8%)

Age (mean, SD)

 10,7 +-2,5

 10,2 +- 2,0

 10,4 +- 2,3

Inpatients aged 6-11 years (n, %)

22 (59,4%)

26 (70,3%)

 48 (64,9%)

Inpatients aged 12-16 years (n, %) 12-16

 15 (40,5%)

11 (29,7%)

 26 (35,1%)

Inpatients with mixed dentition (n, %)

 20 (54,1%)

 26 (70,3%)

 46 (62,2%)

Inpatients with permanent dentitions (n, %)

 17 (45,9%)

 11 (29,7%)

 28 (37,8%)

Revised text:

This table was added to the manuscript in the new section 3.1. Demographic data.

3.1. Demographic Data

Table 1 presents the demographic characteristics age, gender and dention in the study population. All children and adolescents had at least one permanent tooth. 

Table 1. Demographic characteristics of the study population ,intervention group (IG) and control group (CG)

Demographical characteristics

IG

CG

All

Male (N, %)

23

(62.2%)

26

(70.3%)

49

(66.2%)

Female (N, %)

14

(37.8%)

11

(29.7%)

25

(33.8%)

Age in years (Mean ± SD)

10.7

(2.5)

10.2

(2.0)

10.4

(2.3)

In-patients aged 6 - 11 years (N, %)

22

(59.4%)

26

(70.3%)

48

(64.9%)

In-patients aged 12 - 16 years (N, %)

15

(40.5%)

11

(29.7%)

26

(35.1%)

In-patients with mixed dentition (N, %)

20

(54.1%)

26

(70.3%)

46

(62.2%)

In-patients with permanent dentitions (N, %)

17

(45.9%)

11

(29.7%)

28

(37.8%)

  1. Concern of the reviewer:

How could mental compromised/ psychiatric patients that too children would understand or get motivated to the information given through pamphlet?

Our response:

Flyer are very often used to disseminate information. However, we did NOT expect that this flyer would motivate the in-patients towards increasing their oral hygiene. However, we preferred to hand out a flyer instead of doing nothing after collecting the oral health parameter of the control group.

We included in the limitations of the study, that this flyer was identical for all included in-patients, regardless of their age, and not suitable – especially for the younger participants.

                Revised text:

The used information flyer was identical for all age groups, so it may not have been equally appropriate for all in-patients, especially the younger ones.

Reviewer 3 Report

Rating the Manuscript

  • Originality/Novelty: The results provide partial progress because the reliability of the results is questionable considering that the test was conducted among children with mental, motor and various other difficulties. 
    • how is it possible for minor children with mental disabilities to agree to research (materials and methods)
      • How is sit possible to explain tooth brushing technique to patients with mental disorders? There is a lot of controversy in this article
  • Quality of Presentation: the research was conducted among children with mental and motor disorders. Considering the examined groups of children, it is not sufficiently clarified what kind of therapy/medication they are taking, what kind of diet, how well they are physically capable. There are many factors that influence the results obtained. Also, it is not clear how an instructional flyer can help children with mental disorders and how applicable it is. The results are expected, but also controversial. Anamnesis data can significantly influence the results of this research, and therefore their credibility is questionable.

Recommendation

  • Reject

Review Report

  • The most of cited references are not published within the last 5 years (only 2 out of 37 are recent)
  • in some references, the abbreviations of the journal names are not listed, but the full names of the journals

Author Response

General remarks of the authors

We are grateful for the comments of the two reviewers. All comments were carefully considered and incorporated into this new draft of the manuscript. The feedback allowed us to clarify important aspects of the study. We are convinced that the revised manuscript presented herewith is improved over the previous version. This letter will address all comments made in full detail and explain the respective changes we have made to the text.

The information Flyer was uploaded as supplimental material.

Reply to Reviewer 

  1. Concern of the reviewer:

Originality/Novelty:The results provide partial progress because the reliability of the results is questionable considering that the test was conducted among children with mental, motor and various other difficulties. 

Our response:

Our study targeted to evaluate an oral healthcare training in this group of patients with special healthcare needs. It is known, that their oral health is considerable lower than of mentally health children and adolescence.

In childhood and adolesence, the most frequent mental disorders are those included in „Behavioural and emotional disorders with onset ususlly occuring in childhood and adolesence – ICD 10 F90-F98) with an intrinsic heterogeneity. Affective disorders are less frequent, but occur also at young ages,

Children and adolescents with mental health disorders have the potential to learn and train coping strategies for their mental problems and thus improve their own resilience. This learning and empowerment potential might be used to train even such supposedly simple activities like tooth- brushing. Only a fraction of children with mental health disorders also exhibit a reduction in intelligence or limiting motor problems. For this reason, cognitive-practical learning methods are also regularly used in psychiatric-psychological therapy.

Due to the various learning skills of children and adolescents with mental health problems we do think, that our approach to teach those in-patients produced reliable data.

Revised text:

none

  1. Concern of the reviewer:

how is it possible for minor children with mental disabilities to agree to research (materials and methods)

Our response:

Children and adolescents with mental health disorders might express their willingness to do (or not to do) something in a comparable way to those without mental problems. As stated earlier, only few in-patients had reduced intelligence.

We requested written informed consent from all patients and their legal guardians at study enrolment.  Unwillingness to cooperate or to continue participation, expressed by the in-patient, was considered reason to exclude.

Revised text:

none

  1. Concern of the reviewer:

How is it possible to explain tooth brushing technique to patients with mental disorders? There is a lot of controversy in this article

Our response:

Children and adolescents with mental health disorders are capable to learn – tooth brushing as well as much more complex coping strategies, which are trained in Behavioural Therapy (Verhaltenstherapie).

Revised text:

none

  1. Concern of the reviewer:

Quality of Presentation: the research was conducted among children with mental and motor disorders. Considering the examined groups of children, it is not sufficiently clarified what kind of therapy/medication they are taking, what kind of diet, how well they are physically capable. There are many factors that influence the results obtained.

Our response:

Thank you for this remark. We reported the mental health disorders more precisely – including the ICD10 and DSM-IV- Codes. This way we hope to increase clarity regarding the medical condition of the inpatients. In-patients received normal hospital food and none of the in-patients suffered from motor disabilities.

Revised text:

please see Tables 

  1. Concern of the reviewer:

Also, it is not clear how an instructional flyer can help children with mental disorders and how applicable it is.

Our response:

Flyer are very often used to disseminate information. However, we did NOT expect that this flyer would motivate the in-patients towards increasing their oral hygiene. However, we preferred to hand out a flyer instead of doing nothing after collecting the oral health parameter of the control group.

We included in the limitations of the study, that this flyer was identical for all included in-patients, regardless of their age, and not suitable – especially for the younger participants.

                Revised text:

The used information flyer was identical for all age groups, so it may not have been equally appropriate for all in-patients, especially the younger ones.

  1. Concern of the reviewer:

The results are expected, but also controversial. Anamnesis data can significantly influence the results of this research, and therefore their credibility is questionable.

Our response:

Thank you for this remark. Anamnestic data regarding the mental health, family and social surroundings were collected by the psychiatric staff. Although many factors play important roles in oral health – we focused in our study on the effectiveness of oral healthcare-TRAINING on dental plaque reduction by self effectuated tooth brushing in order to narrow the wide range of important imfluences to oral health.

We hope that we succeeded to report more clearly about our research.

  1. Concern of the reviewer:

The most of cited references are not published within the last 5 years (only 2 out of 37 are recent)

Our response:

Thank you for this remark. We searched the literature again and added more recent literature.

Revised text:

Please see: references

  1. Concern of the reviewer:

in some references, the abbreviations of the journal names are not listed, but the full names of the journals

Our response:

The literature was cited in a uniform way, without full names listed and using abbreviation as required by the author guidelines

Revised text:

Please see: References

Reviewer 4 Report

Thank you for the manuscript, which is overall appropriate for the readership of IJERPH and very appropriate for the special issue - Promoting Oral Health for Special Needs Patients. The content of the issue is appropriate and timely and of scientific interest. It highlights that the group of children and adolescents are considered at risk group in terms of dental and oral health. Therefore, these children and adolescents need individual oral health-related prophylaxis and dental care support programs. The manuscript highlights - as an RCT - in which form such offers should be structured to be effective and efficient. The manuscript needs some editing and improvement in a few parts.

Fundamentally, the authors are to be congratulated for this work. Below are some considerations in hopes of helping you improve the manuscript slightly.

1.) Title: ok

2) Introduction: The introduction is overall comprehensible, appropriate in length, and provides a concrete introduction to the need for the study, among other things. I still recommend some concrete A.) epidemiological data for prevalences of psychiatric diseases in children and adolescents from Germany and

B.) oral health-related epidemiological data (caries experience, gingivitis) for children and adolescents with psychiatric diseases from Germany.

3.) Materials:

Please specify here in 2.4 (intervention) -: A.) How long the respective instruction in the IG covered in terms of time and what were the differences in terms of content/methodology for the different age groups? In addition, please insert a statement whether the information flyer for the CG was also identical for children aged 6 years to that for 16 year olds. If there were no differences, I question the extent to which this one flyer was equally appropriate for all ages in the study group (6-16yrs) - or perhaps mention it as a limitation of the study?

Consideration: Include information flyer as a supplement if necessary?

Please delete item 2.6, as it is included in the Part: Institutional Review Board Statement and Informed Consent Statement. Add missing information (number of the German Registry of Clinical Trails) there.

4.) Results: In the presentation of the results, the age of the participants is missing in all tables. It is imperative that this sociodemographic information be added so that a meaningful and adequate classification of the results can be made, among other things, in relation to the oral health-related characteristic data. Overall, the issue of age range needs to be named more visibly in the manuscript. If necessary, corresponding effects in relation to the different age groups need to be named and discussed.

Furthermore, especially in Tables 1 and 2, it is difficult to understand whether the data are coherent with each other.

In table 1, the CG comprises 74 patients - i.e. all patients had at least one permanent teeth?

But only 46 had at least one primary teeth? If this were the case, both groups would comprise different age groups and a comparison of the data - ll.174-178 would only be considered adequate to a limited extent. The continuous text speaks of primary dentition and not of teeth.

It also seems somewhat contradictory that table 2 now speaks of mixed dentition and that the group of permanent dentition now includes only 28 patients. In addition, the data on TI do not seem to be entirely comprehensible. Table 1 - All permanent teeth: 2.3 (0.5) and Table 2: 2. 2 (0.4) - similar for CG and IG.

5.) Discussion: The own results on oral health-related prevalences could be briefly compared here - see above. Note 1.) - be compared with the literature/data inserted in the (new) Introduction. In addition - see above. Note 4.) - discuss the age range.

Further editorial comments:

1.) In the Introduction - ll. 63 and 66 - please remove the blanks (after literacy and approaches).

2) In table 1: in the title please add - before intervention bz. TI 0 - if necessary.

3) In table 3: please add the ICD group designation to the categories (e.g. Neurotic, stress-related and somatoform disorders - ICD. F40-F48?). This will make it clear to the reader which diagnoses are involved in each case.

4) Standardize the citation, e.g. number 1 - extend by the complete indication of the page number 186! This is stored in the rest of the literature list. Also, standardize the journal notation - abbreviation, yes or no? Currently both variants are included. Indicate nr. 29 - when last accessed. Source 34 - please adapt to the style of the other sources.

Author Response

General remarks of the authors

We are grateful for the comments of the two reviewers. All comments were carefully considered and incorporated into this new draft of the manuscript. The feedback allowed us to clarify important aspects of the study. We are convinced that the revised manuscript presented herewith is improved over the previous version. This letter will address all comments made in full detail and explain the respective changes we have made to the text.

The information Flyer was uploaded as supplimental material.

Reply to Reviewer 

Thank you for the manuscript, which is overall appropriate for the readership of IJERPH and very appropriate for the special issue - Promoting Oral Health for Special Needs Patients. The content of the issue is appropriate and timely and of scientific interest. It highlights that the group of children and adolescents are considered at risk group in terms of dental and oral health. Therefore, these children and adolescents need individual oral health-related prophylaxis and dental care support programs. The manuscript highlights - as an RCT - in which form such offers should be structured to be effective and efficient. The manuscript needs some editing and improvement in a few parts.

Fundamentally, the authors are to be congratulated for this work. Below are some considerations in hopes of helping you improve the manuscript slightly.

Our response:

Thank you very much for this appreciation. We are delighted to observe the interest of the reviewer in this group of patients with special healthcare needs

And we would like to thank also for the detailed and specific remarks, which contribute to improve our manuscript.

  1. Concern of the reviewer:
    Title: ok

  1. Concern of the reviewer:

Introduction: The introduction is overall comprehensible, appropriate in length, and provides a concrete introduction to the need for the study, among other things. I still recommend some concrete A.) epidemiological data for prevalences of psychiatric diseases in children and adolescents from Germany and

B.) oral health-related epidemiological data (caries experience, gingivitis) for children and adolescents with psychiatric diseases from Germany.

Our response:

Thank you for this remark. We added recent epidemiological data for prevalence of mental health disorders in Germany. It is difficult to obtain relevant and representative data regarding oral health of children and adolescents with mental health disorders in Germany, since medical and dental diagnoses are not centrally combined. Schmidt et al. request therefore an inter-professional database. This way, oral health of various groups of patients might be analyzed and vulnerable groups detected. [Schmidt, P., Reis, D., Schulte, A. G., & Fricke, O. (2022). Diagnostic Prevalence of Dental Findings in Children, Adolescents and Young Adults with Mental Disorders Compared to Healthy People-an Analysis and Estimation Based on Claims Data from 2019. Psychotherapie, Psychosomatik, Medizinische Psychologie.]

We added to the introduction also data regarding distress of dentist expereinced when treating children with mental health disorders. This ist a major factor contributing tu the inferior oral health in this group.

Revised text:

In Europe, 9.9% of all schoolchildren require mental health care (1). In Germany, published prevalence of psycho-emotional disorders ranges between 8,6% (2), 16.9%(3) and 28% (4).

Patients with mental health disorders might be more susceptible to oral health problems than the general population due to several risk factors: lack of motivation for self-care and oral hygiene, xerostomia caused by psychiatric medications, as well as fear and difficulty in accessing oral healthcare services (1-3). Concomitantly, dentists experience high levels of own psychological distress when treating children or adolescents with psycho-emotional disorders(4)

  1. Concern of the reviewer:

Materials:

Please specify here in 2.4 (intervention) -: A.) How long the respective instruction in the IG covered in terms of time and what were the differences in terms of content/methodology for the different age groups? In addition, please insert a statement whether the information flyer for the CG was also identical for children aged 6 years to that for 16 year olds. If there were no differences, I question the extent to which this one flyer was equally appropriate for all ages in the study group (6-16yrs) - or perhaps mention it as a limitation of the study?

Consideration: Include information flyer as a supplement if necessary?

Please delete item 2.6, as it is included in the Part: Institutional Review Board Statement and Informed Consent Statement. Add missing information (number of the German Registry of Clinical Trails) there.

Our response:

The instruction time in the intervention group took about 10 to 15 minutes depending on the questions of the in-patients. The explanations were adapted in terms of vocabulary, complexity, metaphor while the content was similar, but with individualization. E.g. if the child had difficulties to clean the lower front teeth, than the dentist trained suitable techniques for this region; another child might have problems brushing erupting molars, with swollen gingiva – then the dentist explained that issues and trained also tooth brushing in this specific situation.

Yes, the information flyer for the control group was identical for all children aged 6 years to 16 years. There were no different flyers. It could be possible, that the flyer was not equally appropriate for all ages.  We added this to the limitation section of the study.

Section 2.6. was deleted and the number of the German Registry of Clinical Trials added to the “Institutional Review Board Statement”.

Revised text:

In section 2.4. Intervention:

The time spent on conversation amounted to approximatively 10-15minutes.

All in-patients of the CG received an identical information flyer (36)

In section Strengths and Limitations (297-298)

The used information flyer was identical for all age groups, so it may not have been equally appropriate for all in-patients, especially the younger ones.”

  1. Concern of the reviewer:

Results: In the presentation of the results, the age of the participants is missing in all tables. It is imperative that this sociodemographic information be added so that a meaningful and adequate classification of the results can be made, among other things, in relation to the oral health-related characteristic data. Overall, the issue of age range needs to be named more visibly in the manuscript. If necessary, corresponding effects in relation to the different age groups need to be named and discussed.

Furthermore, especially in Tables 1 and 2, it is difficult to understand whether the data are coherent with each other.

In table 1, the CG comprises 74 patients - i.e. all patients had at least one permanent teeth?

But only 46 had at least one primary teeth? If this were the case, both groups would comprise different age groups and a comparison of the data - ll.174-178 would only be considered adequate to a limited extent. The continuous text speaks of primary dentition and not of teeth.

It also seems somewhat contradictory that table 2 now speaks of mixed dentition and that the group of permanent dentition now includes only 28 patients. In addition, the data on TI do not seem to be entirely comprehensible. Table 1 - All permanent teeth: 2.3 (0.5) and Table 2: 2. 2 (0.4) - similar for CG and IG.

Our response:

Thank you very much for these remarks – as they help to report the data clearer.

The mean age of all in-patients was 10.4 ± 2,3. There was no significant difference between the IG (10.7 ± 2.5) and the CG (10.2 ± 2.0). We added for improved reporting a new table 1 with demographic characteristics of the study population and age groups.

3.1. Demographic Data

Table 1 presents the demographic characteristics age, gender and dention in the study population. All children and adolescents had at least one permanent tooth. 

Table 1. Demographic characteristics of the study population, intervention group (IG) and control group (CG)

Demographical characteristics

IG

CG

All

Male (N, %)

23

(62.2%)

26

(70.3%)

49

(66.2%)

Female (N, %)

14

(37.8%)

11

(29.7%)

25

(33.8%)

Age in years (Mean ± SD)

10.7

(2.5)

10.2

(2.0)

10.4

(2.3)

In-patients aged 6 - 11 years (N, %)

22

(59.4%)

26

(70.3%)

48

(64.9%)

In-patients aged 12 - 16 years (N, %)

15

(40.5%)

11

(29.7%)

26

(35.1%)

In-patients with mixed dentition (N, %)

20

(54.1%)

26

(70.3%)

46

(62.2%)

In-patients with permanent dentitions (N, %)

17

(45.9%)

11

(29.7%)

28

(37.8%)

For data-analysis we decided to compare patients with permanent dentition (older dental age) to those with mixed dentitions (younger dental age). Due to the developmental heterogeneity in patients with mental health disorders, we decided not to split by age - and preferred the dentition. Therefore, the older and younger in-patients were grouped by dentition-status.

Table 1 (new version: table 2) describes the oral health parameters of the primary and permanent teeth of all in-patients and in both groups (IG and CG). We reported Caries, Gingivitis and dental Plaque with focus on the teeth: on primary teeth (within a mixed dentition) and on permanent teeth (within a mixed dentition or within a permanent dentition).

The other tables are focused on the children. For more clarity, we changed headings in the tables.

In-patients

N

Patients

CG/IG

TI at t0

TI at t1a

All

Mean

(SD)

CG

Mean

(SD)

IG

Mean

(SD)

p

All

Mean (SD)

CG

Mean

(SD)

IG

Mean

(SD)

p

All

37 / 37

2.2

(0.5)

2.3

(0.5)

2.1

(0.5)

0.22

2.0

(0.5)

2.1

(0.5)

2.0

(0.5)

0.39

In-patients with mixed dentition

26 / 20

2.2

(0.5)

2.2

(0.5)

2.1

(0.5)

0.35

2.0

(0.4)

2.1

(0.5)

2.0

(0.4)

0.53

In-patients with permanent dentition

11 / 17

2.2

(0.4)

2.3

(0.4)

2.2

(0.5)

0.34

2.0

(0.6)

2.1

(0.6)

2.0

(0.5)

0.56

All patients had at least on permanent tooth – so there were no in-patients with primary dentition.

The TI at t0 of all patients (2.2) in table 2 (complete dentition) is not identical with the TI at t0 of the permanent teeth in table 1 (2.3), here the primary teeth with a better value (1.9) are missing.

  1. Concern of the reviewer:

Discussion: The own results on oral health-related prevalences could be briefly compared here - see above. Note 1.) - be compared with the literature/data inserted in the (new) Introduction. In addition - see above. Note 4.) - discuss the age range.

Our response:

We included the comparison of the prevalences.

Revised text:

This randomized clinical trial investigated the effect of IndOHCT in children and adolescents hospitalized with mental health disorders. The assessed oral health parameters dental caries and gingivitis were considerable higher than those reported in germen children and adolescents with intellectual disabilities (38). This finding emphasises the need for improved oral care in this group of special needs patients.

All elements of the Information-Motivation-Behavioural skills model were implemented and techniques of motivational interviewing (29) applied. Care was taken to communicate and act age-related, since major differences in cognitive and motor skills in the age range from 6 – 16 years were to respect.

  1. Concern of the reviewer:

Further editorial comments:

1.) In the Introduction - ll. 63 and 66 - please remove the blanks (after literacy and approaches).

2) In table 1: in the title please add - before intervention bz. TI 0 - if necessary.

3) In table 3: please add the ICD group designation to the categories (e.g. Neurotic, stress-related and somatoform disorders - ICD. F40-F48?). This will make it clear to the reader which diagnoses are involved in each case.

4) Standardize the citation, e.g. number 1 - extend by the complete indication of the page number 186! This is stored in the rest of the literature list. Also, standardize the journal notation - abbreviation, yes or no? Currently both variants are included. Indicate nr. 29 - when last accessed. Source 34 - please adapt to the style of the other sources.

Our response:

  • thank you for the overlooked blanks, they were removed
  • the heading of the table was changed
  • ICD 10 and DSM-IV Codes were added in table 3 (new:4) and also 4 (new:5)
  • Citation was standardized generally according to the authors guideline

Revised text:

                2) Table 1. Oral health in child and adolescent psychiatric in-patients at the beginning (t0).

Psychiatric Diagnosis, Medication

N

Patients

CG/IG

TI at t0

TI at t1a

All

Mean

(SD)

CG

Mean

(SD)

IG

Mean

(SD)

p

All

Mean (SD)

CG

Mean

(SD)

IG

Mean

(SD)

p

Neurotic, stress-related and somatoform disorders
(ICD 10 - F40-F48)

13 / 13

2.2

(0.5)

2.3

(0.5)

2.2

(0.5)

0.36

2.0

(0.5)

2.0

(0.5)

2.0

(0.4)

0.86

Behavioural and emotional disorders with onset usually occurring in childhood and adolescence
(ICD 10 - F90-F98)

33 / 31

2.2

(0.5)

2.3

(0.5)

2.2

(0.5)

0.41

2.1

(0.4)

2.1

(0.5)

2.0

(0.4)

0.32

Abnormal intrafamilial relations
(DSM-IV Axis 5 - 1)

11 / 14

2.3

(0.4)

2.3

(0.3)

2.2

(0.5)

0.47

2.0

(0.6)

2.0

(0.5)

2.0

(0.6)

0.72

Mental disorder, deviant behaviour or disability in the family

(DSM-IV Axis 5 - 2)

24 / 20

2.2

(0.5)

2.3

(0.5)

2.1

(0.5)

0.15

2.0

(0.5)

2.1

(0.5)

2.0

(0.4)

0.47

Abnormal educational Conditions
(DSM-IV Axis 5 - 4)

20 / 26

2.3

(0.4)

2.4

(0.4)

2.2

(0.4)

0.05*

2.2

(0.4)

2.2

(0.4)

2.1

(0.1)

0.20

Abnormal immediate surroundings
(DSM-IV Axis 5 - 5)

30 / 30

2.2

(0.5)

2.3

(0.5)

2.1

(0.5)

0.23

2.0

(0.5)

2.1

(0.5)

2.0

(0.4)

0.31

Acute, stressful life events
(DSM-IV Axis 5 - 6)

12 / 10

2.3

(0.4)

2.3

(0.3)

2.3

(0.4)

0.85

2.0

(0.4)

2.0

(0.4)

2.1

(0.4)

0.65

Attention deficit hyperactivity disorder (ADHD) medication

9 / 11

2.4

(0.5)

2.4

(0.6)

2.3

(0.4)

0.57

2.0

(0.4)

2.2

(0.5)

1.9

(0.3)

0.19

Antipsychotic drug medication

7 / 5

2.3

(0.6)

2.3

(0.6)

2.3

(0.6)

0.88

2.2

(0.5)

2.4

(0.5)

2.0

(0.2)

0.18

Round 2

Reviewer 2 Report

Pls replace capital 'N' with a small 'n' and capital 'P' with small 'p' in the tables.

Rest all seems fine.

Author Response

Thank you for this observation - we replaced N and P accordingly - and the overall favourable vote.

Reviewer 3 Report

I thank the authors for their respectful remarks.

What is still not clearly clarified are the different learning potentials, different perception and acquisition of knowledge between certain groups of children with special needs. Emotional disorders and mental health disorders require different research approaches and criteria. Although the number of patients with reduced intelligence is small, it is impossible to ignore this fact and treat all children in an identical way. The severity of the disorder significantly determines cognitive abilities.

If other reviewers have accepted the work in this format, I agree with its publication. In that case, you can consider these remarks as instructions for some upgrade of this research in future work.

I am thankful on respected remarks and additional interventions in the article.

Author Response

Respected Reviewer,

we do understand very well your concern regarding the intelectual capacity and learning potentials of our cohort. And indeed, in case of intelectual impairment, each and every training concept has to take such limitations into account with sensibility. Nevertheles, the co-authors from the Psychiatric clinic, both experienced psychiatric doctors, did diagnostics also on axis II of the DSM V (intelectual and personality disorders). During clinical data extraction from the medical records and interprofessional data analysis, no concern was risen regarding intelectual impairment of the included patients . 

Further research focussing on patients with intelectual and/or learning impairment could reveal new and interesting aspects.

We appreciate your concerns and thank you for your favourable decision.

Reviewer 4 Report

Dear authors' team, thank you very much for consistent and adequate implementation of all mentioned hints. From my point of view, the manuscript its very suitable for publication in the present form. Thus, it only remains for me to congratulate the authors' team on this paper.

Author Response

Dear reviewer, we are grateful for your appreciation and the favourable vote. Thank you very much.